# Preparation and Physicochemical Analysis of *Camellia sinensis* cv. ‘Ziyan’ Anthocyanin Microcapsules

**DOI:** 10.3390/foods13040618

**Published:** 2024-02-19

**Authors:** Ruixin Xue, Xiang Yuan, Hong Jiang, Hong Huang, Xiaocong Luo, Pinwu Li

**Affiliations:** College of Horticulture, Sichuan Agricultural University, Number 211 Huimin Road, Chengdu 611130, China; xueruixin121@163.com (R.X.); ermu919@163.com (X.Y.); jhong667@163.com (H.J.); 15196483268@163.com (H.H.); 18200328981@163.com (X.L.)

**Keywords:** *Camellia sinensis* cv. ‘Ziyan’, anthocyanin, microencapsulation, fourier transform infrared spectroscopy, differential scanning calorimetry, thermogravimetric analysis

## Abstract

The new tea cultivar Ziyan has a high content of anthocyanin and ester catechins in the raw material, but the conventional processing and application methods are limited. To explore its application potential, the freeze-drying method was used to prepare microcapsules with an embedding time of 30 min, solid content of 30%, and core to wall ratio of 1:10 (g/g). The anthocyanin recovery was 95.94 ± 0.50%, and the encapsulation efficiency was 96.15 ± 0.11%. The stability of microcapsules and composite wall materials was evaluated in the simulation system. Results showed that microcapsules employing a maltodextrin–gum arabic ratio of 2:8 (*w*/*w*) as the wall material significantly reduced degradation rates, extending anthocyanin half-life under various storage conditions. Characterization indicated improved physical properties of Ziyan anthocyanin powder post-microencapsulation. FT-IR and DSC- revealed the formation of a new phase between anthocyanins and wall materials, leading to increased enthalpy and enhanced thermal stability. The microencapsulation results of this experiment proved that the storage stability of anthocyanin was effectively enhanced.

## 1. Introduction

Ziyan (ZY) is a new tea variety with purple buds obtained by single-plant selection and systematic breeding in the medium- and small-leaf group of Sichuan, which is a shrub type, medium-leaf type, and late-blooming species. The buds, leaves, and stems of ZY have a dark purple color. The anthocyanin content of ZY is about 0.88 mg/g [1]. Anthocyanin is a class of glycosylated polyphenolic compounds consisting of the basic anthocyanin skeleton coupled with a sugar or acyl group, and more than 600 species of anthocyanidins have been identified in nature [2].

However, there are some difficulties in using anthocyanins as natural colorants in the food industry, and their special structure leads to easy degradation of anthocyanin under the action of internal and external factors. Most of the degradation patterns follow the first-order kinetic mode [3], while factors, such as pH, temperature, light, and storage, have the greatest influence on stability [4]. In addition, the stability of microcapsules can also be affected by oxygen, metal ions, enzymes, and some other food components [5].

Microencapsulation is defined as a technology that encapsulates a bioactive substance as a capsule core with an encapsulant (wall or shell material) [6]. The selection of suitable wall materials is the most important step in the microencapsulation process, and some materials have been shown to have good protective effects. Commonly used wall materials include natural gums (gum arabic, sodium alginate, carrageenan, etc.) [7], proteins (whey proteins, soy proteins, etc.) [8], carbohydrates (maltodextrins and cellulose derivatives) [9], or lipids (waxes, emulsifiers) [10]. Seke et al. [11] found that the stability and bioavailability of bioactive components in red fruits (such as strawberries) can effectively be improved by microencapsulation. According to Berg et al. [12], microcapsules with the addition of highly esterified citrus pectin and beet pectin based on maltodextrin-encapsulated blueberry anthocyanin resulted in higher hydration capacity and better retarding of anthocyanin. Similarly, Moser P [13] used different concentrations and ratios of soy protein, maltodextrin, and whey protein/maltodextrin mixtures to encapsulate grape juice by spray drying. The results show that microencapsulation can better reduce powder hygroscopicity, viscosity, and total color difference during storage. Burin et al. [2] evaluated the stability of grape anthocyanin microcapsules (MAC) in a soft drink system. The results show that the combination of maltodextrin and gum arabic exhibits the longest half-life and the lowest degradation constant among all wall combinations. Similarly, according to Wyspianska et al. [14], microcapsules prepared with maltodextrin and inulin as wall materials were able to eliminate the undesirable odor of the core material and improve its stability in beverage systems, and there was a gain in the health effects of such active microcapsules for traditional beverages. Therefore, the simulation of application scenarios for anthocyanin microcapsules is necessary to achieve the pigmentation of natural anthocyanins. In addition, the freeze-drying technique is one of the best methods for drying unstable compounds containing anthocyanin during microcapsule preparation [15]. Gong et al. [16] used a combination of maltodextrin (MD) and whey protein isolate (WPI) to spray-dry strawberry puree. The results showed that MD-WPI increased the powder recovery efficiency by 56%. The low temperature and vacuum conditions reduce the risk of undesirable changes, resulting in anthocyanin microcapsules with higher color values and better activity [17].

In this work, we investigated the preparation of Ziyan anthocyanin (ZYAC) microcapsules for the first time. The performances of Maltodextrin, Gum Arabic, and Soy Protein Isolate were compared as microcapsule wall materials, and the preparation process was also optimized to enhance the recovery of ZYAC during processing. In addition, the optimized anthocyanin microcapsules were characterized by SEM electron microscopy, Fourier infrared spectroscopy, and differential scanning calorimetry. We evaluated the performance of the microcapsules in extending the storage period of ZYAC and improving the stability of anthocyanins.

## 2. Materials and Methods

### 2.1. Materials

The experimental material was Ziyan tea fresh leaves from Mingshan District, Ya’an City, Sichuan Province, and the material tested was one bud and three leaves. Stability test materials, mainly including sodium chloride, potassium chloride, calcium chloride, magnesium chloride, aluminum chloride, zinc sulfate, copper sulfate, and sodium acetate were purchased from Chengdu Kelong Chemical Reagent Factory (Chengdu, China). Wall materials mainly included maltodextrin, gum arabic, and soybean isolate protein, and were purchased from Xilong Chemical Co., Ltd. (Chengdu, China). All other chemicals were of analytical purity and purchased from Chengdu Kelong Reagent Factory. Distilled water was used throughout the experiments.

### 2.2. Preparation of ZYAC Powder

ZY fresh leaves (one bud three leaves) were picked. According to the evaporation of solid sample method (103 ± 2 °C) to prepare dry samples, samples were adequately passed through the grinder mill (FW80 grinder mill, Shanghai YETO Technology Co., Shanghai, China) and a 60 mesh sieve and were then stored at 4 °C in sealed storage for later use. A total of 0.1 g of ZY powder was weighed and extracted by sonication in a 65 °C water bath for 39 min at 50 W (KH5200E ultrasonic cleaner, Kunshan Wo Chuang Ultrasonic Instrument Co., Kunshan, China); it was then centrifuged at 4500 rpm for 15 min and the supernatant was stored at 4 °C, protected from light. A total of 3.0 g of DM 301 resin was added to 50 mL of 0.2 mg/mL ZYAC extract and placed in a thermostatic oscillator, protected from light, and shaken at 100 rpm for 6 h. Saturated resin was formulated to absorb surface water; 1.0 g was weighed, and 25 g of ethanol solution was added at pH = 1.0. A total of 25 mL of 70% ethanol solution was desorbed at 100 rpm with gentle shaking for 6 h. After resin purification, the anthocyanin eluent was collected, and the eluent was concentrated and evaporated using vacuum rotary evaporation (LC-RE-2000E rotary evaporation, Shanghai Li-Chen Instrument Technology Co., Shanghai, China) until there was no ethanol in the eluent, and then the concentrate was extracted and concentrated with ethyl acetate, and this step was repeated five times. The aqueous layer of the obtained extract was taken and placed on the vacuum rotary evaporator, and the residual ethyl acetate in the concentrate was evaporated cleanly, and then it was freeze-dried (LGJ-18 freeze dryer, North South Instruments Ltd., Zhengzhou, China) for 24 h at −50 °C to obtain ZYAC powder.

### 2.3. Screening of Wall Materials for MAC

Three wall materials were selected for single-wall embedding and combined wall embedding, which include maltodextrin (MD)with DE = 20, gum arabic (GA), and soy protein isolate (SPI). All the wall materials were completely solubilized in distilled water at 45 °C, cooled to room temperature, and stored at 4 °C overnight to complete hydration. The composite wall materials were prepared in the ratios of 9:1, 8:2, 7:3, 6:4, 5:5, 4:6, 3:7, 2:8, and 1:9 (g/g), and all the wall materials were added with ZYAC in the ratio of a core-to-wall ratio of 1:10 to make the total solids content of the solution 20%. They were then mixed in a high-speed homogenizer for 30 min at 100 rpm with a magnetic stirrer and at 10,000 rpm under vacuum.

### 2.4. Determination of the Embedding Rate of MAC

To evaluate the effect of microencapsulation, refer to the method of Ab Rashid et al. [9] with minor modifications. A total of 0.1 g of ZYAC microencapsulated lyophilized powder was accurately weighed in a grinding bowl, and 1mL of distilled water and 10mL of anhydrous ethanol were added to extract the total anthocyanins. After extraction, the samples were centrifuged at 5000 rpm for 10 min, the extracts were passed through a 0.45 μm filtration membrane, and the anthocyanin content was determined. Alternatively, 0.1 g of ZYAC microencapsulated lyophilized powder was weighed to determine the surface anthocyanin content; briefly, the sample was placed in a centrifuge tube, 10 mL of anhydrous ethanol was added, and vortex extraction was carried out for 30 s. The extract was centrifuged at 5000 rpm for 10 min, and the extract was passed through a 0.45 μm filter membrane. The surface anthocyanin content was determined. The rate of microencapsulation was calculated according to the following equation:Embedding rate = (M_(TAC)_ − M_(SAC)_/M_(TAC)_)(1)
where M_(TAC)_ is the total anthocyanin content of microcapsules, mg; M_(SAC)_ is the surface anthocyanin content, mg.

### 2.5. Stability Testing of MAC

The degradation kinetics test during storage was carried out by referring to the method of Burin et al. [2] with slight modifications. A simulated soft drink system (containing 4% sucrose, 0.4% citric acid, 0.1% sodium citrate, 0.15% dextrose, 0.12% NaCl, and 0.1% potassium sorbate) was established. Three composite wall MACs containing equal amounts of anthocyanins and ZYAC powder were added to this simulated beverage system. The groups were stored in four different situations to examine the effect of the environment on MAC, respectively. To assess the degradation rate, anthocyanin content was measured at 7 d intervals for a total of 6 cycles, and the degradation kinetic equation was fitted using the obtained anthocyanin content. The rate constants for the degradation reactions were calculated using the kinetic equations for the zero, primary, and secondary reactions, respectively. The half-life t_1/2_ was calculated to assess the degradation effect [18], and the equation was calculated as follows:
Zero-level reaction kinetics: C_t_ = C_0_ − kt (2)
(3)$$Primary\;reaction\;kinetics:\ln\left(\frac{C_{t}}{C_{0}}\right)=-\mathrm{kt}$$
Second-order reaction kinetics: ln C_t_ = ln C_0_ − kt (4)
(5)$$t_{1/2}=\frac{\ln\;2}{k}$$
where C_t_ is the detected anthocyanin content at moment t, mg/g, C_0_ is the initial anthocyanin content, mg/g, k is the reaction rate constant, and t_1/2_ is the half-life.

### 2.6. Optimization of the Preparation Process of MAC

MD and GA were used as wall materials, and the ratio of composite wall materials was 2:8 for the embedding of ZYAC. After embedding, the microcapsules were homogenized with a high-speed homogenizer at 10,000 rpm for 10 min, and then vacuum freeze-dried for 24 h. The microcapsules were then stored in a brown sealed bottle at −20 °C. The freeze-dried microcapsule powder was collected and stored in a brown sealed bottle at −20 °C. The results of the test were indexed by the recovery rate of anthocyanin, and the recovery rate was calculated as follows:
Recovery rate R (%) = (M_(TAC)_ − M_(SAC)_)/M_(AAC)_(6)
where M_(TAC)_ is the total anthocyanin content of microcapsules, mg; M_(SAC)_ is the surface anthocyanin content, mg; M_(AAC)_ is the anthocyanin content of the total input, mg.

Three indicators (total solids, microcapsule core-to-wall ratio, and embedding time) were fixed in the ZYAC microcapsules, and the recovery of anthocyanins was analyzed by one-way experiments. On the basis of the one-way experiments, the L9(34) orthogonal test was carried out to determine the optimal process parameters for the preparation of microcapsules by taking the recovery of ZYAC as the evaluation index. The design of the factors and levels of the orthogonal test is shown in Table 1.

### 2.7. Microencapsulation Property Testing

#### 2.7.1. Detection of Physical Properties of MAC

A suitable amount of powder before and after embedding was weighed and dried to constant weight in a constant temperature blast dryer at 105 °C, and the mass of powder before and after drying was recorded. Referring to the method of Tonon et al. [19] with slight modifications, the sample powder (0.2 g) before and after embedding was placed in a desiccator for 7 d under light protected conditions. The mass of the sample before and after storage was weighed and hygroscopicity was expressed as the amount of water absorbed per 100 g of dry matter (g/100 g). Referring to the method of de Souza [20], 0.1 g of each anthocyanin sample before and after embedding was weighed, 50 mL of distilled water was added, and the sample was magnetically stirred at 100 rpm for 30 min. The mixed solution was transferred to a centrifuge tube and centrifuged at 3500 rpm for 5 min. A total of 25 mL of the supernatant was transferred to a petri dish and dried in a drying oven at 105 °C until the constant weight and solubility were calculated.

Referring to the method of Odabaş and Koca [21], 2.0 g of the powder sample was weighed into a 10 mL cylinder and the bulk density (g/mL) was calculated by determining the ratio of sample mass to volume. Vibrational density (g/mL) was determined by tapping the cylinder to make the volume constant and calculating the ratio of the final read volume to the sample mass. The flowability and cohesion of the microcapsules were evaluated by the Carr index (CI) and Hausner ratio (HR), respectively, which were calculated using the following equations:CI = (Vibrated density − bulk density)/vibrated actual density(7)
HR = Vibration density/packing density(8)

#### 2.7.2. The Morphology of MAC

For scanning electron microscopy observation of microcapsule morphology, a small amount of lyophilized anthocyanin microcapsule powder was spread flat on double-sided adhesive tape. The residual powder was blown off, and the sample was sprayed with gold color and kept under vacuum for 5 min. The powder was analyzed with a scanning electron microscope at 10 KV with a magnification of 500–2000 times.

#### 2.7.3. Fourier Infrared Spectroscopy Analysis

Microcapsules of MAC, ZYAC, MD, GA, and CK were analyzed by FTIR spectroscopy. Five samples were examined by co-milling KBr with the samples and placing them in a tablet press with a set wavelength range of 4000–500 cm^−1^ and a scan number of 32/64.

#### 2.7.4. Thermal Stability Analysis

Measurements were taken to study the change in the heat absorption of particles before and after anthocyanin embedding by using the simultaneous thermal analyzer. The thermal properties were checked over the temperature range of 25 °C to 300 °C by placing approximately 5 mg of sample in an aluminum tray, purging with nitrogen as an inert gas, and heating at a rate of 10 °C per minute.

#### 2.7.5. Test Equipment

The model and manufacturer information of the instruments used in Section 2.7.2, Section 2.7.3 and Section 2.7.4 are mentioned in Table 2.

### 2.8. Data Processing

Data were processed and analyzed using SPSS 8.0.6; all images were plotted using Origin 2020. A one-way ANOVA test was used to determine significant differences between means using Duncan’s multiple comparisons test, and all test results were averaged over three trials and expressed as mean ± standard deviation.

## 3. Results and Discussion

### 3.1. Wall Screening Results of MAC

#### 3.1.1. Comparison of MAC Embedding Rates

The single-wall material SPI (91.19 ± 0.32%) showed the highest encapsulation of ZYAC, whereas there was no significant difference in the encapsulation effect of MD and GA when they were used as wall materials alone. MD has a low film formation rate, and the microencapsulation structure may be affected when it is used alone. As for GA, as a compound polysaccharide compound with a highly disproportionate structure, its encapsulation effect will be affected by its physical properties, and the encapsulation efficiency of the two is relatively low.

Usually, a single carrier does not have good encapsulation ability, so the use of carrier mixtures is a more effective way to protect compounds. In this experiment, significant difference analysis was employed, and the results showed that the encapsulation efficiency was improved by the composite wall material. As shown in Table 3, the highest encapsulation rate of 97.13 ± 0.87% was achieved when the composite ratio of MD and GA was 2:8, which was improved by 6.5% and 14.9% compared to the single-wall materials MD and GA, respectively. The encapsulation rate of 98.01 ± 0.78% was achieved when the composite ratio of MD and SPI was 8:2, which was, respectively, improved by 14.6% and 7.5% compared to the single-wall materials MD or SPI. SPI and GA obtained a maximum embedding rate of 97.92 ± 0.33% at a composite ratio of 2:8, which was 7.4% and 13.6% higher than a single SPI or GA, respectively. The difference in wall embedding rate was determined by the interaction between wall and anthocyanin molecules during the embedding process, and a study by Shahidi Noghabi and Molaveisi [22] confirmed that the complexation of MD and GA was able to reduce the water content of the loaded chlorophyll microcapsules and enhance the embedding rate (77.19%), which is similar to the present experimental results. This was because, with the increase in GA ratio, more short-chain hydrophilic groups were brought in, and its emulsification property was improved, the film-forming and permeability were enhanced, the embedding rate was increased [23], and the aggregation of GA by MD was improved as well. When MD was compounded with SPI, the two were able to form a dense and complete honeycomb structure through hydrogen bonding, which could enhance the encapsulation efficiency; however, when the ratio of SPI was too high, the liquid viscosity was too large, and the embedding rate was decreased [24]. Similar results have also been shown in studies on the microencapsulation of flavonoids from Cornus officinalis [25].

In addition, the composite wall embedding rates of SPI and GA were not significantly different at all compounding ratios provided in this experiment, which may be attributed to the fact that the incorporation of GA improves protein aggregation, resulting in consistently high embedding rates (>95%) and less differentiation. Mansour et al. [26] used the SPI composite GA in the study of red raspberry anthocyanin microencapsulation and obtained embedding rates ranging from 93.05% to 98.87%, similar to the results of the present experiments. The team also revealed the existence of interactions between the wall and the core, which is one of the reasons for the high rates of embedding, using the FT-IR test.

The embedding efficiencies of the six wall materials were ranked as MD + SPI a, SPI + GA a, MD + GA a, SPI b, MD c, and GA c. The composite wall materials with appropriate composite ratios did not have any significant difference, all of them were greater than 95%, the embedding effect was good, and almost all of the cores were sufficiently encapsulated; therefore, the subsequent experiments selected the composite ratios of MD + GA 2:8 (g/g) for M1, MD + SPI 8:2 (g/g) for M2, and SPI + GA 2:8 (g/g) for M3 to further prepare Zi Yan anthranilic acid microcapsules. Therefore, in the subsequent experiments, the MD + GA composite ratio of 2:8 (g/g) was chosen as M1, the MD + SPI composite ratio of 8:2 (g/g) was chosen as M2, and the SPI + GA composite ratio of 2:8 (g/g) was chosen as M3, and the MACs were prepared for the further comparative stability study.

#### 3.1.2. Results of Degradation Kinetics Tests

Four storage conditions were set up and three reaction kinetic models were fitted to describe the degradation of anthocyanins before and after encapsulation (Table 4) The storage resistance of the three wall materials was compared by examining the changes in anthocyanin concentration in the soft drink system during 42 days of storage.

In Figure 1, the degradation process and degradation kinetics of ZYAC and its three microcapsules in a beverage system are fitted, which shows that the stability of anthocyanin is significantly improved by the three microcapsules compared with unembedded anthocyanin under all storage conditions.

The longest half-life is obtained for M1 under shading at 4 °C, which is 2.46-fold higher than that of anthocyanin, and M1 combined with low temperature are the ideal conditions for storing ZYAC. However, it is worth noting that the accumulation and precipitation of anthocyanin are observed during the storage period, which may be due to the decrease in solute solubilization under low temperature. It will have some effect on the product clarification, and the concentration of anthocyanin needs to be adjusted in the commercial application. In addition, since the calculated half-life is much longer than the storage period set in the test, the 4 °C storage test can be further expanded to obtain more accurate data.

At 37 °C, M2 effectively prevented the degradation of anthocyanin. On the one hand, the amine group and carbonyl group of the protein formed hydrogen bonds with the hydrophilic region of anthocyanins. On the other hand, the hydrophobic region of the protein interacted with the benzene ring of anthocyanin, and the addition of MD prevented the self-aggregation of the protein, which further enhanced the stabilizing effect [27], which was the same as that of the combined wall of SPI and GA (M3). As for M1 microcapsules, the incorporation of GA resulted in improved film-forming properties of MD for better trapping and encapsulation of anthocyanin molecules, while the complex formed by anthocyanins and GA through hydrophobic interactions [28] made the yellow fused ions less susceptible to hydrophilic attack by water molecules, and the stability was significantly improved.

Light is another unfavorable factor that triggers degradation. For UV conditions, the degradation rate of anthocyanin increases from 0.0154 to 0.0301, and the half-life is shortened to 20.09 d. The final retention rates of the three microcapsules were M1 (72.35%) > M2 (58.39%) > M3 (43.98%), among which M1 was effective in improving the photostability of chlorogenic glycosides. The degradation rate was reduced to 0.0132, and the t1/2 was 52.67 d, which greatly prolonged the half-life of anthocyanins.

### 3.2. Optimization of the Preparation Process of Anthocyanin Microcapsules

As shown in Figure 2, the results of one-way experiments showed that within 30 min, the shorter the embedding time, the higher the surface anthocyanin content, and the lower the recovery, but the recovery of anthocyanins decreased after the embedding time exceeded 30 min, so 30 min was chosen as the center point. The increase in recovery was no longer apparent above 25% solids, suggesting that the thickening effect may be optimal at around 25% solids. The best results were obtained at a core-to-wall ratio of 1:8 (g/g).

The tests were conducted according to the orthogonal test design table and the results obtained are shown in Table 5 and Table 6. The optimal combination was an embedding time of 30 min, a solid content of 30%, and a core-to-wall ratio of 1:10 (g/g). Based on this optimization, three replicate validation experiments were carried out. The recovery of the MAC was measured to be 95.94 ± 0.50%, and the embedding rate at this time was 96.15 ± 0.11%, which indicated that the recovery of the anthocyanin, the embedding effect, and the properties of the prepared microcapsules required further testing to clarify their physicochemical parameters.

### 3.3. Physicochemical Characterization of Microcapsules

Before and after microencapsulation, the difference in water content change in ZYAC was not significant, but microencapsulation increases the solubility of anthocyanin by 16.80%. Microencapsulation reduced the CI and HR of ZYAC, as shown in Table 7. Cohesion, flowability, and the quality of the microencapsulated powder were also improved. However, the microcapsule had a higher hygroscopicity, which could be caused by GA as a hydrophilic colloid and could also be related to the method of freeze-drying.

In Figure 3, the change in color before and after encapsulation originates from the change in pH due to the introduction of the wall material, while the change in brightness originates from the dilution effect of the wall material. It was observed that the powder of the microcapsules is more delicate and slightly more agglomerated, which might be related to the increase in hygroscopicity in the aforementioned assay results.

SEM micrographs of the particle morphology of the MAC were shown in Figure 4. The microcapsules had an irregular surface morphology similar to that of porous crushed glass, which was a typical lyophilized powder morphology, and the interior of the microcapsules showed an obvious honeycomb structure, which explains its high embedding rate and high solubility. Due to the formation of pores, the anthocyanin molecules were embedded in them. As a result, little anthocyanin remains on the surface and the embedding rate was increased. The honeycomb structure led to an increased contact area with water molecules, while the agglomeration effect between glycosides was eliminated, resulting in a reduced tendency to agglomerate in liquid matrices, and solubility was enhanced.

### 3.4. Result of Fourier Infrared Spectroscopy Analysis

As shown in Figure 5, the characteristic broad bands around 3200 cm^−1^ for each sample are mainly related to the O-H stretching vibration of water molecules bound in the samples. The bands from 2800 to 3000 cm^−1^ may be due to the asymmetric or symmetric stretching of methyl groups. ZYAC shows peaks in the dense bands around 800 to 1500 cm^−1^ (fingerprint region), from 1029 to 1692 cm^−1^, where C-O stretching is observed at 1239 cm^−1^, and at 1692 cm^−1^, which is attributed to C-O-C bending vibration. The bands at 1289 cm^−1^ correspond to the aromatic ring of the flavonoids and the backbone of the C-O-C groups’ stretching vibration, the peak band at 1447 cm^−1^ is attributed to the C-N vibration, 1520 to 1600 cm^−1^ is related to the typical C=C vibration in the aromatic ring [29], and 1604 cm^−1^ is attributed to the C=O vibration of the aromatic ring functional group of the anthocyanin, the benzopyran ring. The broad absorption bands in the 3600 to 3100 cm^−1^ regions are also related to the stretching of the OH moiety in the flavonoids or phenolic acids vibrations, further proving that the prepared sample powders are characterized by anthocyanin.

In this experiment, the hydroxyl group shifts from 3271 cm^−1^ to 3305 cm^−1^ as compared to the unloaded microcapsules. The vibrational telescoping peaks here may correspond to the hydrogen bonding, while the absorption peak at the fingerprint region of 1029 cm^−1^ is similarly red-shifted and the vibrational peak strengthens, corresponding to the formation of hydrogen bonding, and the hydroxyl peak is stabilized. In addition, compared with ZYAC, 2930 cm^−1^ is where the C-H bond of maltodextrin extends and covers the peaks of polyphenolic compounds in the extract. The spectral bands from 1029 to 1692 cm^−1^ are covered by the wall material with weakened amplitudes, which shows that the core and wall material are effectively bonded together.

### 3.5. Results of DSC-TGA

The encapsulation effect of the wall and core materials can trigger the absence and movement of the heat absorption peaks, leading to changes in the melting, boiling, or sublimation points of the crystals, as shown in Figure 6a, which indicates that the heat absorption phenomenon changed before and after the encapsulation of ZYAC. The first heat absorption peaks appear between 47.38 °C and 84.88 °C for microcapsules and between 52.29 °C and 77.29 °C for anthocyanin, which could be caused by thermal desorption of volatiles and water. There is also a flat new absorption peak between 284.88 °C and 369.88 °C for microcapsules, which means anthocyanin molecules are successfully introduced into the microcapsules and form an inclusion.

Figure 6b shows the temperature profiles obtained from the TGA test, from which it can be seen that the quality of microcapsules fluctuates less in the range of 100 °C to 200 °C, which can satisfy the requirements of most food sterilization and processing treatments. The peak temperature of the DTG curve represents the temperature corresponding to the maximum rate of weight loss. The temperature of the peak is 257.38 °C, and the peak of the microcapsule is 307.38 °C, which makes microcapsules more stable under the high-temperature processing. Therefore, the microcapsules are more stable under high-temperature processing. In addition, the enthalpy of ZYAC powder is 16.79 J/g, while that of microcapsule powder is 26.16 J/g, which proves that the energy required for the phase transition of microcapsules is higher and that the embedding effect significantly improves the thermal stability of anthocyanin, which provides favorable conditions for the processing of ZY.

MAC is produced by M1 microcapsules with MD-GA as the composite wall material under the conditions of an embedding time of 30 min, solid content of 30%, and core-to-wall ratio of 1:10 (g/g), which provide the best protection for anthocyanin. The recovery of anthocyanin in the final product is 95.94 ± 0.50% and the embedding rate is 96.15 ± 0.11%. The microencapsulation process reduces the water content and improves the solubility. SEM electron microscopy observes that the powder is honeycomb-shaped, and DSC/TGA and FTIR analysis show that ZYAC microcapsules form a new physical phase. The experimental results enhance the stability of ZYAC and provide a basis for expanding its application.

## 4. Conclusions

In this study, unitary or binary composites of MD, GA, and SPI were used as wall materials for embedding ZYAC and process optimization for microcapsule preparation, and the properties of MAC were also characterized. The best protection of anthocyanins was achieved under the condition of MD + GA: 2:8 (g/g) as wall material, and the half-life of ZYAC was prolonged with a 2.01-fold enhancement. MD + SPI: 8:2 (g/g) and SPI + GA: 2:8 (g/g) only enhanced the half-life of ZYAC by 1.43- and 1.34-fold, respectively. Orthogonal tests revealed that an encapsulation time of 30 min, a solid content of 30%, and a core-to-wall ratio of 1:10 (g/g) were the optimal parameters for the preparation of M1, under which the anthocyanin recovery was 95.94 ± 0.50% and the encapsulation rate was 96.15 ± 0.11%. Microencapsulation reduced the water content from 4.64% to 4.29% and increased the solubility from 80.89% to 94.48%. In addition, in this study, MAC was observed to be honeycomb-shaped using SEM electron microscopy, and FTIR and DSC/TGA analyses showed that MAC formed a new physical phase with increased enthalpy and enhanced thermal stability of anthocyanosides. The present study was effective in enhancing the stability of the storage of anthocyanin microcapsules.

## Figures and Tables

**Figure 1 foods-13-00618-f001:**
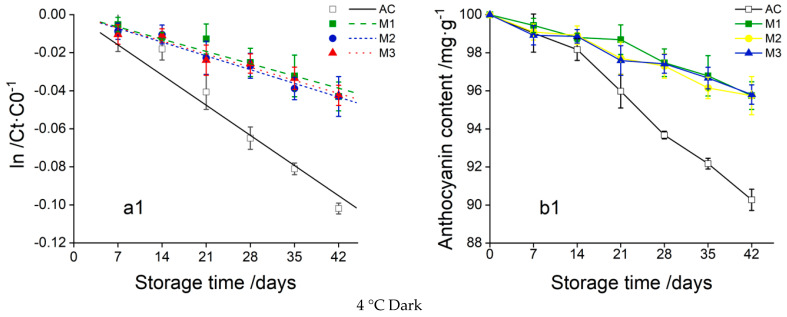
Linear fitting plot of primary degradation kinetics of anthocyanins under different storage conditions (**a1**–**a4**); changes in anthocyanin content with time (**b1**–**b4**); AC is Ziyan anthocyanin; M1 is MD + GA (2:8); M2 is MD + SPI (8:2); M3 is SPI + GA (2:8).

**Figure 2 foods-13-00618-f002:**
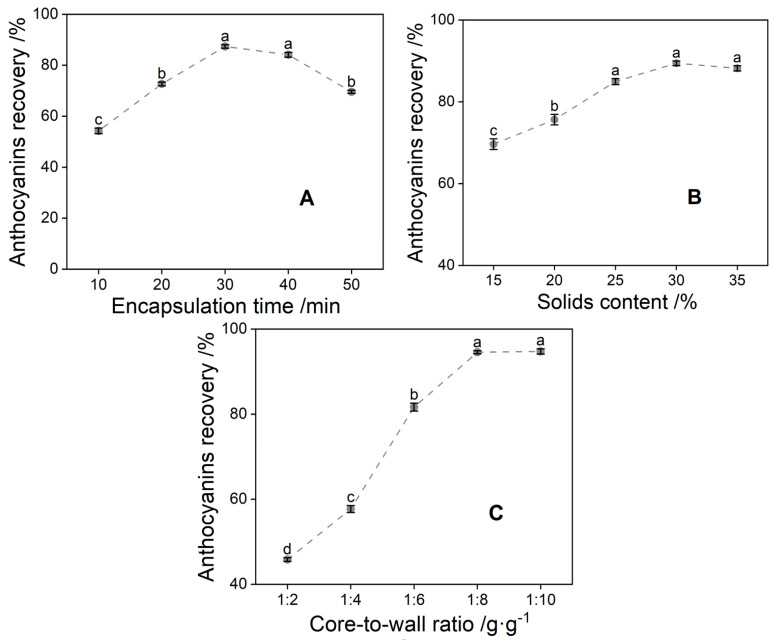
Influence factors of anthocyanin recovery ((**A**) encapsulation time; (**B**) solids content; (**C**) core-to-wall ratio); Significant differences within groups are indicated using letters. There are significant differences between different letters, significance level *p* < 0.05.

**Figure 3 foods-13-00618-f003:**
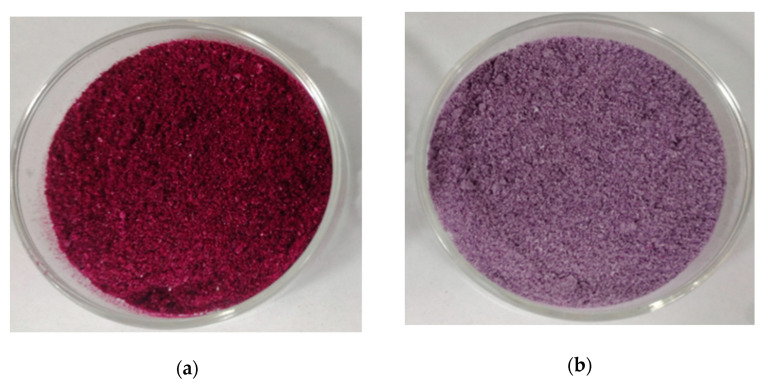
Anthocyanin powder of Ziyan (**a**); microcapsule powder of anthocyanins in Ziyan (**b**).

**Figure 4 foods-13-00618-f004:**
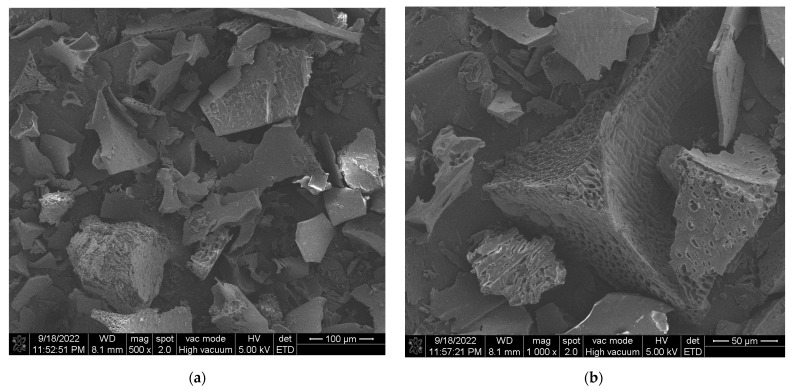
SEM of microcapsules: magnification 500 (**a**); 1000 (**b**); 2000 (**c**); 5000 (**d**).

**Figure 5 foods-13-00618-f005:**
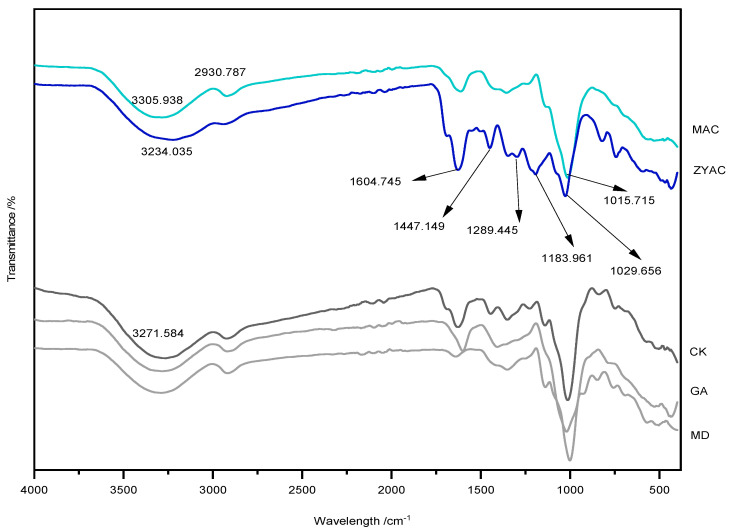
Fourier infrared spectroscopy of different materials: microcapsule (MAC); Ziyan anthocyanins (ZYAC); no-load microcapsules (CK); gum arabic (GA); maltodextrin (MD).

**Figure 6 foods-13-00618-f006:**
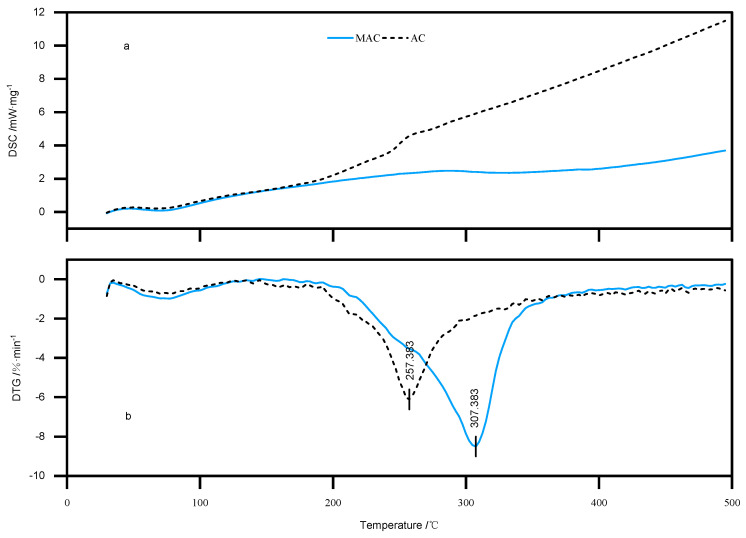
Thermogravimetric and differential thermal analysis plot of anthocyanins and microcapsule; (**a**) thermogravimetric result of nthocyanins and microcapsule, (**b**) differential thermal analysis result of anthocyanins and microcapsule.

**Table 1 foods-13-00618-t001:** Orthogonal test factors and horizontal test table.

Considerations	Level (of Achievement etc.)		
	1	2	3
A—Encapsulation time (min)	20	30	40
B—Solids content (%)	25	30	35
C—Core-to-wall ratio (g/g)	6	8	10

**Table 2 foods-13-00618-t002:** Test instruments.

Serial No.	Instrument Name	Model and Manufacturer
1	Field-emission scanning electron microscope (FESEM)	FEI-NOVASEM 230; FEI Company (Hillsboro, OR, USA)
2	High speed homogenizer	FSH-2A; Shanghai Yao Technology Development Co. (Shanghai, China)
3	Fourier infrared spectrometer	iS5; Thermo Fisher Scientific (Waltham, MA, USA)
4	Synchronized thermal analyzer	STA 449 F5; NETZSCH Scientific Instruments Trading Ltd. (Shanghai, China)

**Table 3 foods-13-00618-t003:** Effect of composite wall material ratio on encapsulation efficiency.

Ratio of Wall Materials	Encapsulation Efficiency		
	MD + GA	MD + SPI	SPI + GA
9:1	85.56 ± 1.02 e	96.31 ± 1.14 ab	96.55 ± 1.28 a
8:2	89.5 ± 1.28 d	98.01 ± 0.78 a	95.61 ± 0.97 a
7:3	93.3 ± 1.43 bc	96.77 ± 0.99 ab	96.16 ± 1.04 a
6:4	91.29 ± 0.88 cd	92.8 ± 1.05 bc	95.22 ± 1.15 a
5:5	94.74 ± 0.61 ab	96.37 ± 0.78 ab	95.78 ± 0.59 a
4:6	95.45 ± 0.72 ab	90.92 ± 0.86 cd	96.96 ± 0.61 a
3:7	96.26 ± 0.33 ab	87.78 ± 1.42 d	97.71 ± 0.85 a
2:8	97.13 ± 0.87 a	80.48 ± 1.72 e	97.92 ± 0.33 a
1:9	97.13 ± 0.32 a	76.69 ± 1.09 e	95.26 ± 1.43 a

Significant differences within groups are indicated using letters. There are significant differences between different letters, significance level *p* < 0.05.

**Table 4 foods-13-00618-t004:** Degradation kinetic equations and half-life of anthocyanins and microcapsules under different storage conditions.

Storage Conditions	Sample	Zero-Level Reaction Kinetics C_t_ = C_0_ − kt	Primary Reaction Kinetics ln (C_t_/C_0_) = −kt	Second-Order Reaction Kinetics 1/C_t_ = 1/C_0_ + kt	Half-Life t_1/2_ (d)
4 °C Dark	AC	C_t_ = 0.01 − 0.0015 t R^2^ = 0.9551	ln (C_t_/C_0_) = −0.0023 t R^2^ = 0.9501	1/C_t_ = 100 + 0.1656 t R^2^ = 0.9447	305.4 d
	M1	C_t_ = 0.01 − 0.0084 t R^2^ = 0.9565	ln (C_t_/C_0_) = −0.0009 t R^2^ = 0.9364	1/C_t_ = 100 + 1.428 t R^2^ = 0.9630	753.0 a
	M2	C_t_ = 0.01 − 0.0133 t R^2^ = 0.7774	ln (C_t_/C_0_) = −0.0010 t R^2^ = 0.9692	1/C_t_ = 100 + 3.683 t R^2^ = 0.9379	673.0 c
	M3	C_t_ = 0.01 − 0.0148 t R^2^ = 0.9447	ln (C_t_/C_0_) = −0.0010 t R^2^ = 0.9534	1/C_t_ = 100 + 5.624 t R^2^ = 0.8488	702.4 b
25 °C Dark	AC	C_t_ = 0.01 − 0.0047 t R^2^ = 0.9503	ln (C_t_/C_0_) = −0.0154 t R^2^ = 0.9755	1/C_t_ = 100 + 0.0657 t R^2^ = 0.93377	44.92 d
	M1	C_t_ = 0.01 − 0.0006 t R^2^ = 0.9394	ln (C_t_/C_0_) = −0.0077 t R^2^ = 0.9493	1/C_t_ = 100 + 0.6160 t R^2^ = 0.9437	90.49 a
	M2	C_t_ = 0.01 − 0.0075 t R^2^ = 0.9587	ln (C_t_/C_0_) = −0.0091 t R^2^ = 0.9498	1/C_t_ = 100 + 1.163 t R^2^ = 0.9826	76.59 b
	M3	C_t_ = 0.01 − 0.0085 t R^2^ = 0.9523	ln (C_t_/C_0_) = −0.0106 t R^2^ = 0.9550	1/C_t_ = 100 + 1.460 t R^2^ = 0.9679	65.51 c
25 °C Ultraviolet	AC	C_t_ = 0.01 − 0.0007 t R^2^ = 0.9698	ln (C_t_/C_0_) = −0.0301 t R^2^ = 0.9871	1/C_t_ = 100 + 0.0737 t R^2^ = 0.9687	23.09 c
	M1	C_t_ = 0.01 − 0.0054 t R^2^ = 0.9538	ln (C_t_/C_0_) = −0.0132 t R^2^ = 0.9891	1/C_t_ = 100 + 0.7472 t R^2^ = 0.9397	52.67 a
	M2	C_t_ = 0.01 − 0.0103 t R^2^ = 0.9267	ln (C_t_/C_0_) = −0.0202 t R^2^ = 0.9832	1/C_t_ = 100 + 2.041 t R^2^ = 0.9724	34.27 b
	M3	C_t_ = 0.01 − 0.0083 t R^2^ = 0.9871	ln (C_t_/C_0_) = −0.0224 t R^2^ = 0.9897	1/C_t_ = 100 + 1.415 t R^2^ = 0.9123	31.01 b
37 °C Dark	AC	C_t_ = 0.01 − 0.0007 t R^2^ = 0.9523	ln (C_t_/C_0_) = −0.0375 t R^2^ = 0.9784	1/C_t_ = 100 + 0.0704 t R^2^ = 0.9538	18.49 c
	M1	C_t_ = 0.01 − 0.0062 t R^2^ = 0.9693	ln (C_t_/C_0_) = −0.0157 t R^2^ = 0.9918	1/C_t_ = 100 + 0.9018 t R^2^ = 0.9305	44.15 a
	M2	C_t_ = 0.01 − 0.0065 t R^2^ = 0.9534	ln (C_t_/C_0_) = −0.0152 t R^2^ = 0.9718	1/C_t_ = 100 + 2.355 t R^2^ = 0.9656	45.63 a
	M3	C_t_ = 0.01 − 0.068 t R^2^ = 0.9235	ln (C_t_/C_0_) = −0.0208 t R^2^ = 0.9889	1/C_t_ = 100 + 2.123 t R^2^ = 0.9647	33.37 b

AC is Ziyan anthocyanin; M1 is MD + GA (2:8); M2 is MD + SPI (8:2); M3 is SPI + GA (2:8); Significant differences within groups are indicated using letters. There are significant differences between different letters, significance level *p* < 0.05.

**Table 5 foods-13-00618-t005:** Orthogonal test protocol and results.

Test No.	A—Encapsulation Time	B—Solids Content	C—Core-to-Wall Ratio	Recovery (%)
1	1 (20)	1 (25)	1 (6)	72.68 ± 0.16
2	1	2 (30)	2 (8)	80.56 ± 0.57
3	1	3 (35)	3 (10)	79.12 ± 0.32
4	2 (30)	1	2	93.55 ± 1.05
5	2	2	3	95.58 ± 0.71
6	2	3	1	88.34 ± 0.98
7	3 (40)	1	3	81.20 ± 0.46
8	3	2	1	79.25 ± 0.78
9	3	3	2	79.93 ± 0.68
K_1_	232.36	247.43	240.29	
K_2_	277.49	255.39	254.04	
K_3_	240.38	247.41	255.90	
k_1_	77.45	82.48	80.10	
k_2_	92.50	85.13	84.68	
k_3_	80.13	82.47	85.30	
R	15.04	2.66	5.20	

K_i_ represents the sum of the corresponding experimental results when the level number on any column is i; k_i_ denotes the mean; R indicates range.

**Table 6 foods-13-00618-t006:** The results of ANOVA that show the influence of each factor.

Origin	SS	Df	MS	F	*p*
A	386.4655	2	193.2327	166.598	0.006 **
B	14.1158	2	7.0579	6.085	0.141
C	48.4660	2	24.2330	20.893	0.046 *
e	2.3198	2	1.1599		

* Represents a significant difference, *p* ≤ 0.05; ** represents a highly significant difference, *p* ≤ 0.01.

**Table 7 foods-13-00618-t007:** Physical properties of anthocyanins before and after microencapsulation.

Physical Property	Anthocyanin	Microencapsulation
Packing density (g/cm^3^)	0.40 ± 0.01 a	0.42 ± 0.01 a
Solidification density (g/cm^3^)	0.50 ± 0.01 a	0.48 ± 0.01 a
Carr index (CI)	20.38 ± 1.54 a	13.08 ± 1.37 b
Hausner ratios (HR)	1.26 ± 0.02 a	1.15 ± 0.02 b
Moisture content (%)	4.64 ± 0.30 a	4.29 ± 0.18 a
Solubility (%)	80.89 ± 2.55 b	94.48 ± 1.11 a
Absorbance (g/100 g)	10.98 ± 0.30 b	13.27 ± 0.55 a

Significant differences within groups are indicated using letters, significance level *p* < 0.05.

## Data Availability

The original contributions presented in the study are included in the article, further inquiries can be directed to the corresponding author.

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
