# Peer review of "Preparation and Physicochemical Analysis of Camellia sinensis cv. ‘Ziyan’ Anthocyanin Microcapsules"

_foods, 2024, doi:10.3390/foods13040618_

Round 1

Reviewer 1 Report

Comments and Suggestions for Authors

The topic of the manuscript is relevant and provides interesting insights for its readers. However, there is room for improvement in the following aspects:

1.     Use passive voice consistently throughout the manuscript.  

2.     Update references. Use more references after 2018. Additionally, the manuscript lacks sufficient references. It would be beneficial to include more references overall.

3.     The full title of chapter 3 is missing. There is also discussion, not only results.

4.     Enhance the quality of Figures 1 and 2, as they suffer from poor quality Adjustments are needed to improve readability and comprehension.

5.     Introduction: Clearly highlight the main aim and novelty of the study.

Author Response

Cover Letter

Manuscript ID: foods-2869348

Dear Assistant Editor Danika Zhang and Reviewer:

We feel great thanks for your professional review work on our article. As you are concerned, there are several problems that need to be addressed. According to your nice suggestions, we have made extensive corrections to our previous draft. The reviewer comments are laid out below in italicized font and specific concerns have been numbered. Our response is given in normal font and changes to the manuscript are highlighted within the document by using yellow-colored text. Point-by-point responses to the nice editor and three nice reviewers are listed below this letter.

Comments and Suggestions for Authors:

Reviewer #1

Comments 1: Use passive voice consistently throughout the manuscript. 

Response 1: Thank you for your kind reminder. We have converted the full manuscript to passive voice.

Comments 2: Update references. Use more references after 2018. Additionally, the manuscript lacks sufficient references. It would be beneficial to include more references overall.

Response 2: We sincerely appreciate the valuable comments. We have checked the literature carefully and added more references from 2018 onwards in the revised manuscript.

Comments 3:  The full title of chapter 3 is missing. There is also discussion, not only results.

Response 3: We think that is a good point. Therefore, we have changed the title of chapter 3 from " Results " to " Results and Discussion "( line 209).

Comments 4: Enhance the quality of Figures 1 and 2, as they suffer from poor quality Adjustments are needed to improve readability and comprehension. 

Response 4: Thank you for your comments. We apologize for the lack of clarity in Figure 1 and Figure 2, which we have improved to make them easier to read.

Comments 5:  Introduction: Clearly highlight the main aim and novelty of the study.

Response 5: Thank you for your kind reminder. We apologize for the ambiguity of the introduction, and we have placed it in the revised manuscript so that it more clearly describes our purpose and novelty (line 68-75).

According to the Assistant Editor and Reviewers’ comments, we tried our best to improve the manuscript and made extensive modifications to our manuscript. All revisions to the manuscript have been highlighted. Thank you again for your positive comments and valuable suggestions to improve the quality of our manuscript.

Thank you very much for your attention and time. We are looking forward to hearing from you.

Sincerely,

Pinwu Li

Feb 15, 2024

Reviewer 2 Report

Comments and Suggestions for Authors

The presented manuscript “Preparation and Physicochemical Analysis of Ziyan Anthocyanin Microcapsules” is a good exercise of anthocyanin from Ziyan tea microcapsules and physicochemical analysis. The manuscript is not correctly presented, its need improves the redaction. The following comment is in order to improve the manuscript.

 Tittle.

Please, add the scientific name to tea ziyan.

 Abstract section.

Please, add the conditions and method for obtain the microcapsules.

Introduction section.

Please, check the space in lines 37,41, 44 and 50.

Line 58, please, add the meaning of ZYAC.

Section 2.1.

Please, rewrite the lines 61 to 62, the text is not clear.

Add the equivalent dextrose grade of the maltodextrin used.

Section 2.2.

Please, describe the “evaporation of solid sample method” in lines 71 and 72. What was the sample temperature?

For revolutions per minutes in the centrifuge conditions please, use “rpm”, lines 74, 77 and 78. Also, In the rest of manuscript.

Section 2.4.

Please, check the description of equation 1 in lines 106 to 107.

Section 2.6.

Please, check the description of equation 6 in lines 128 and 129. The variable “M(AAC) is not mentioned.

In Table 1, please add the unites for variables considerate.

Section 2.7.4.

Please, add the equipment used for thermal analysis.

Section 3.1.1.

The results obtained with singled walled material are not presented. On line 179 it is mentioned that there is information in Table 2, however, only the results for the composite wall material are shown.

In the table 2 the meaning of letter is not mentioned, also the statistical method used, must be mentioned.

Line 186, table 3 does not present the mentioned results, check the missing table.

Lines 185 to 200. According with the authors the composite wall material for MD+SPI present the highest encapsulation rate. Which is not significant with another composites wall material. Please, mention the statistical method used.

In the table 2 the effect to SPI in the composites wall material is interesting, when the mix is SPI+GA, an effect to addition of SPI is not observed. However, when is mixed with MD (MD+SPI), the encapsulation efficiency increases when the SPI concentration reduces. Please, increase the discussion about the SPI wall material.

Section 3.1.2.

Table 3, why used acitretin as reference? The technical information of acitretin is not mentioned in section 2.1. Also, please add the meaning of letters presented in the column 6 (t1/2).

The table 3 is not mentioned in the text.

Figure 1, the axis title in the graphical is not clear, please, increase the font size.

In the axis title correct the unites (mg).

Check redaction in line 229.

Section 3.2.

Please, add the equation used for optimization process by orthogonal test.

Figure 2, the axis title in the graphical is not clear, please, increase the font size.

The figure 2 is not mentioned in the text.

Also, please check the table 5 and 6, figure 5 is not mentioned in the text.

Section 3.3.

Table 6, please correct the unites for volume.

Lines 270 to 274, a characterization by L, a, b, improve the discussion about color.

Section 3.3.

Please, correct the united cm-1.

Figure 5, why the analysis by IR for SPI is not presented?

Conclusion section.

The section presents a summary of the results, please improve it, to present the highlights and main aspects of the research.

References.

Approximately the 40% of references are before that 2014, please update the references.

Author Response

Manuscript ID: foods-2869348

Dear Assistant Editor Danika Zhang and Reviewer:

We feel great thanks for your professional review work on our article. As you are concerned, there are several problems that need to be addressed. According to your nice suggestions, we have made extensive corrections to our previous draft. The reviewer comments are laid out below in italicized font and specific concerns have been numbered. Our response is given in normal font and changes to the manuscript are highlighted within the document by using yellow-colored text. Point-by-point responses to the nice editor and three nice reviewers are listed below this letter.

Comments and Suggestions for Authors:

Reviewer #2:

Comments 1: Tittle. Please, add the scientific name to tea ziyan.

Response 1: Thank you for your kind reminder. We have added Ziyan's scientific name (Camellia Sinensis cv. ‘Ziyan’) to the title (line 2).

Comments 2: Abstract section. Please, add the conditions and method for obtain the microcapsules.

Response 2: Thank you for your comment. We have added the conditions and methods for obtaining microcapsules of Ziyan Anthocyanin into the abstract (line 9-20).

Comments 3: Introduction section. Please, check the space in lines 37,41, 44 and 50. Line 58, please, add the meaning of ZYAC.

Response 3: We feel sorry for our carelessness. Based on your comments, We have examined the spaces at lines 37, 41, 44, and 50(line 36, 46, 49, 52)and corrected them, as well as interpreted the meaning of ZYAC at line 58 and corrected it in the manuscript( line 68).

Comments 4: Section 2.1. Please, rewrite the lines 61 to 62, the text is not clear. Add the equivalent dextrose grade of the maltodextrin used.

Response 4: We are really sorry for our careless mistakes. We have reworded the ambiguity (lines 79-80).

Comments 5: Section 2.2. Please, describe the “evaporation of solid sample method” in lines 71 and 72. What was the sample temperature? For revolutions per minutes in the centrifuge conditions please, use “rpm”, lines 74, 77 and 78. Also, In the rest of manuscript.

Response 5: We sincerely thank the reviewer for careful reading. We have checked and confirmed the experimental temperature of A and added it to the manuscript (line 89-90; Also, we have replaced all r/min with rpm in the article).

Comments 6: Section 2.4. Please, check the description of equation 1 in lines 106 to 107.

Response 6: We are really sorry for our careless mistakes. We have checked the description of equation 1 and modified it (line 130-132).

Comments 7: Section 2.6. Please, check the description of equation 6 in lines 128 and 129. The variable “M(AAC) is not mentioned.

In Table 1, please add the unites for variables considerate.

Response 7: Thank you very much for your careful checks. We apologize for the lack of description, and we have added the elaboration of M(AAC) (line 155-157); also, the appropriate units have been added to con in Table I (line 164).

Comments 8: Section 2.7.4. Please, add the equipment used for thermal analysis.

Response 8: Thank you very much for the valuable. We have added to the text the instruments used for thermal analysis (line 196), and the instrument information combination was edited to section 2.7.5.

Comments 9: Section 3.1.1.

The results obtained with singled walled material are not presented. On line 179 it is mentioned that there is information in Table 2, however, only the results for the composite wall material are shown.

In the table 2 the meaning of letter is not mentioned, also the statistical method used, must be mentioned.

Line 186, table 3 does not present the mentioned results, check the missing table.

Lines 185 to 200. According with the authors the composite wall material for MD+SPI present the highest encapsulation rate. Which is not significant with another composites wall material. Please, mention the statistical method used.

In the table 2 the effect to SPI in the composites wall material is interesting, when the mix is SPI+GA, an effect to addition of SPI is not observed. However, when is mixed with MD (MD+SPI), the encapsulation efficiency increases when the SPI concentration reduces. Please, increase the discussion about the SPI wall material.

Response 9: We are really sorry for our careless mistaken. We incorrectly referenced the tables, we have adjusted the table references to be correct and added comments (line 219-220); for the issue of insignificant encapsulation rate between different composites, we used significant difference analysis to analyze the optimal ratio of the composites, while the differences between different composites were used to determine the optimal composites through subsequent degradation kinetics experiments(line 222-227); meanwhile, we have further analyzed for the issue of the significant effect of SPI on the encapsulation rate when it is combined with MD, but the non-significant effect of SPI on the encapsulation rate when it is combined with GA(line 236-263).

Comments 10: Section 3.1.2. Table 3, why used acitretin as reference? The technical information of acitretin is not mentioned in section 2.1. Also, please add the meaning of letters presented in the column 6 (t1/2).

The table 3 is not mentioned in the text.

Figure 1, the axis title in the graphical is not clear, please, increase the font size.

In the axis title correct the unites (mg).

Check redaction in line 229.

Response 10: We feel sorry for our carelessne. The AC in Table 3. is not ferulic acid, but is short for un-microencapsulated Ziyan Anthocyanin, and we have labeled that item in the notes to the table (line 271); have also labeled the meaning of column 6 (t1/2) (line 269-270); have pointed out the Table 3 reference in the text (line 266); have corrected the problem with the units of the axes in Fig. 1 and adjusted the font size (line 285-290); and checked the problem with 229 (line291).

Comments 11: Please, add the equation used for optimization process by orthogonal test.

Figure 2, the axis title in the graphical is not clear, please, increase the font size.

The figure 2 is not mentioned in the text.

Also, please check the table 5 and 6, figure 5 is not mentioned in the text.

Response 11: Thank you for your careful checks. We have resized the axes captions in Figure 2 (line 314-315) and pointed out in the text where Figures 2 (line 309) and 5(line 359), as well as table 6 and 6, are referenced (line 318; line 335-356).

Comments 12: Section 3.3. Table 6, please correct the unites for volume.

Lines 270 to 274, a characterization by L, a, b, improve the discussion about color.

Response 12: Thank you very much for your reminders. We have modified the units in Table 7(line 339); we are very sorry that further discussion of L,a,b characterization of the colors is not possible at this time due to the condition of the laboratory equipment.。

Comments 13: Section 3.3. Please, correct the united cm-1. Figure 5, why the analysis by IR for SPI is not presented?

Response 13: Thank you very much for your careful checks. We have modified the units (line 359-380); There is no SPI because the optimal embedded wall combination of GA+MD has been determined in the preliminary stability analysis, therefore no IR analysis was performed for SPI.

Comments 14: Conclusion section.The section presents a summary of the results, please improve it, to present the highlights and main aspects of the research.

Response 14: Thank you very much for your suggestion. We have re-written this part and added the Conclusions section according to the Reviewer'

Comments 15: References.

Approximately the 40% of references are before that 2014, please update the references.

Response 15: We think this is an excellent suggestion. We have checked the literature carefully and added more references from 2018 onwards in the revised manuscript.

According to the Assistant Editor and Reviewer’ comments, we tried our best to improve the manuscript and made extensive modifications to our manuscript. All revisions to the manuscript have been highlighted. Thank you again for your positive comments and valuable suggestions to improve the quality of our manuscript.

Thank you very much for your attention and time. We are looking forward to hearing from you.

Sincerely,

Pinwu Li

Feb 15, 2024

Reviewer 3 Report

Comments and Suggestions for Authors

The authors optimized the microencapsulation process to improve the recovery of anthocyanin of
Ziyan. The scope of the work coincides with the topic of the journal. The references are up-to-date,
and the study's proposal is interesting.
The corrections suggested for the manuscript are:
- The summary lacks information about the results of the optimization performed (time, solid content,
ratio).
- The keyword section can be improved, since some words of the manuscript´s title are repeated, I
suggest replacing these words with other keywords to improve the scope of your research in the
scientific website databases.
- In line 31 there should be [2].
- Line 41 is missing a space after the word effects.
- Line 41 should read Berg et al.
- Line 50 should read Burin et al.
- The introduction lacks references to the literature or information about the lack of available literature
reports on the microencapsulation of anthocyanins from Ziyan. It can be assumed that conducting
such research is the authors' original contribution, but it was not indicated in the work and, in my
opinion, it should be supplemented in the introduction.
- Line 58 uses the abbreviation ZYAC but does not provide its expansion.
- In section 2.1, it is worth indicating the exact region where the leaves come from.
- Section 2.2 lacks information on how many ml of solvent was used during extraction and the model
and manufacturer of the grinder mill and ultrasonic cleaner. It is worth specifying the pressure at which
the freeze-dried was performed.
- Line 95 should read Ab Rashid et al.
- Line 110 should read Burin et al.
- In units 2, 3, 4 and 5 there is no explanation for C0, Ct, kt, k.
- Line 119 there is no space after the word effect.
- Line 142 should read Tonon et al.
- Line 146 there is a reference to footnote no. 14. You should check it with references, because there is
an error - a different name.
- Line 151 should read OdabaÅŸ and Koca.
- Line 158 should read 2.7.2. The (…).
- In points 2.7.2, 2.7.3 and 2.7.4, please provide the exact model and manufacturer of the devices on
which individual analyzes were performed.
- The materials and methods section does not describe how the statistical analysis of the results was
carried out and does not provide the name of the software that was used for optimization.
- In the results section, in line 178, the authors write about the lack of statistically significant
differences. Tables 2 and 3 have different letters and no explanation, making verification impossible.
- On line 190 it should be Shahidi Noghabi and Molaveisi.
- In lines 196, 232, 236, 296 there is no space before the footnote.
- In table 3, the superscript at R2 should be corrected.
- In line 249 it says: ''A core-to-wall ratio of 1:8 (g/g) is found 249 to be optimal'', while in line 255 the
ratio is 1:10. This needs to be verified.
- No explanations K1 - r in table 4.
- No explanation * and ** in table 5.
- There is no superscript cm-1 in section 3.4.
- The conclusions need improvement as it should indicate which parameters of the encapsulation
process are optimal.

Author Response

Cover Letter

Manuscript ID: foods-2869348

Dear Assistant Editor Danika Zhang and Reviewer:

We feel great thanks for your professional review work on our article. As you are concerned, there are several problems that need to be addressed. According to your nice suggestions, we have made extensive corrections to our previous draft. The reviewer comments are laid out below in italicized font and specific concerns have been numbered. Our response is given in normal font and changes to the manuscript are highlighted within the document by using yellow-colored text. Point-by-point responses to the nice editor and three nice reviewers are listed below this letter.

Comments and Suggestions for Authors:

Reviewer #3:

Comments 1: - The summary lacks information about the results of the optimization performed (time, solid content, ratio).

Response 1: We think this is an excellent suggestion. We have re-written this part and added the results of the optimization performed for obtaining microcapsules of Ziyan Anthocyanin into the abstract (line 9-20).

Comments 2: - The keyword section can be improved since some words of the manuscript´s title are repeated, I suggest replacing these words with other keywords to improve the scope of your research in the scientific website databases.

Response 2: Thank you for your kind reminder. We've reworked the keywords based on the suggestions (line 21-22).

Comments 3: - In line 31 there should be [2].

Response 3: Thank you for your careful reading. We have corrected the error (line 31).

Comments 4: - Line 41 is missing a space after the word effects. - Line 41 should read Berg et al.
- Line 50 should read Burin et al.

Response 4: We feel sorry for our carelessness. In our resubmitted manuscript, the error is revised. Thanks for your correction (line 39, 42, 43, 46, 52).

Comments 5: - The introduction lacks references to the literature or information about the lack of available literature
reports on the microencapsulation of anthocyanins from Ziyan. It can be assumed that conducting
such research is the authors' original contribution, but it was not indicated in the work and, in my
opinion, it should be supplemented in the introduction.

Response 5: We sincerely appreciate the valuable comments. We have checked the literature carefully and added more references into the introduction part in the revised manuscript(line 38-75).

Comments 6: - Line 58 uses the abbreviation ZYAC but does not provide its expansion.

Response 6: We sincerely thanl the reviewer for careful reading. As suggested by the reviewer, we have provided the expansion of ZYAC (line 68).

Comments 7: - In section 2.1, it is worth indicating the exact region where the leaves come from.

Response 7: As suggested by the reviewer, we have added the exact region where the leaves come from (line 79-80).

Comments 8:  Section 2.2 lacks information on how many ml of solvent was used during extraction and the model and manufacturer of the grinder mill and ultrasonic cleaner. It is worth specifying the pressure at which the freeze-dried was performed.

Response 8: We thank the reviewer for raising these important points. We have added instrument information to the line and expanded further to explain the freeze-drying process (line 90-107).

Comments 9: - Line 95 should read Ab Rashid et al.- Line 110 should read Burin et al.

Response 9: We would like to thank the reviewer for pointing out those issues. We have revised those issues in the manuscript (line 119; line 135).

Comments 10: - In units 2, 3, 4 and 5 there is no explanation for C0, Ct, kt, k.

Response 10: Thank you for the helpful comments. We have re-added to the manuscript all the interpretations of C0, Ct, kt, k (line 146-147).

Comments 11: - Line 119 there is no space after the word effect. - Line 142 should read Tonon et al.

Response 11: We feel sorry for our carelessness. In our resubmitted manuscript, the error is revised. Thanks for your correction (line 144; line 169).

Comments 12: - Line 146 there is a reference to footnote no. 14. You should check it with references, because there is an error - a different name.

Response 12: We are really sorry for our careless mistaken. We have replaced the correct references in the manuscript (line 173).

Comments 13: - Line 151 should read OdabaÅŸ and Koca. - Line 158 should read 2.7.2. The (…).

Response 13: We are really sorry for our careless mistaken. We have replaced the correct references in the manuscript (line 179; line 186).

Comments 14: - In points 2.7.2, 2.7.3 and 2.7.4, please provide the exact model and manufacturer of the devices onwhich individual analyzes were performed.

Response 14: Thank you for your helpful comments. We have presented the information about the instruments used in a table in section 2.7.5 (line 203).

Comments 15: - The materials and methods section does not describe how the statistical analysis of the results wascarried out and does not provide the name of the software that was used for optimization.

Response 15: We thank the reviewer for the positive and constructive comments regarding our paper. Based on the reviewer' comments, we have added the statistical methods and the software used to section 2.8(line 204-208).

Comments 16: - In the results section, in line 178, the authors write about the lack of statistically significant differences. Tables 2 and 3 have different letters and no explanation, making verification impossible.

Response 16: Thank you for kindly reminding us. We wrote that there were no significant differences between the effects of a single wall material on the microcapsules because a single wall material did not improve the performance of the microcapsules much, which is why subsequent experimental attempts were made to combine composite wall materials, with a focus on studying the effect of composite wall materials as being significantly better than a single wall material (line 212-263); In the meantime, we have added the notes to Tables 2. and Table 3. to follow up the tables (line 220, 271).

Comments 17: - On line 190 it should be Shahidi Noghabi and Molaveisi. - In lines 196, 232, 236, 296 there is no space before the footnote. - In table 3, the superscript at R2 should be corrected.

Response 17: Thank you very much for your valuable comments, we have made the appropriate changes in response to the issues you raised (line 233; line 238, 295, 299, 367).

Comments 18: - In line 249 it says: ''A core-to-wall ratio of 1:8 (g/g) is found 249 to be optimal'', while in line 255 the ratio is 1:10. This needs to be verified.

Response 18: We would like to thank the reviewer for pointing out this issue. ''A core-to-wall ratio of 1:8 (g/g) is found 249 to be optimal'' at this time for the one-factor control of the embedding time as well as the total solids unchanged, the core-to-wall ratio was tested, and at this time 1:8 was the optimal ratio, so 1:8 was chosen as the center point for orthogonal experiments. At the later stage, it was said that "a core-to-wall ratio of 1:10 (g/g)" was the optimal choice under the combined condition, because after the orthogonal experiment, it was found that the effect of embedding time on the recovery of anthocyanin was extremely significant, the effect of core-to-wall ratio on the recovery of anthocyanin was significant, and the effect of the content of total solid was not significant on the recovery of anthocyanin, and the effect of three factors on the recovery of anthocyanin was not significant. The effect of the three factors was in the order of A>C>B, and the optimal combination was 30 min of embedding time, 30% of solids content, and 1:10 core-to-wall ratio (g/g). We have made certain linguistic additions here that are easier to understand.

Comments 19: - No explanations K1 - r in table 4. - No explanation * and ** in table 5.

Response 19: We think this is an excellent suggestion. We have annotated the exegesis at the back of the table (line 327-328; line 330-331)

Comments 20: - There is no superscript cm-1 in section 3.4.
Response 20: We sincerely thank the reviewer for careful reading. We have revised the issue in the manuscript (line 358-380).

Comments 21: - The conclusions need improvement as it should indicate which parameters of the encapsulation process are optimal.
Response 21: Thank you very much for your valuable comments, we have made the appropriate changes in response to the issues you raised (line 417-432).

According to the Assistant Editor and Reviewer’ comments, we tried our best to improve the manuscript and made extensive modifications to our manuscript. All revisions to the manuscript have been highlighted. Thank you again for your positive comments and valuable suggestions to improve the quality of our manuscript.

Thank you very much for your attention and time. We are looking forward to hearing from you.

Sincerely,

Pinwu Li

Feb 15, 2024